# Physiologically Based Pharmacokinetic Modeling of Antibiotics in Children: Perspectives on Model-Informed Precision Dosing

**DOI:** 10.3390/antibiotics14060541

**Published:** 2025-05-24

**Authors:** Ryota Tanaka, Kei Irie, Tomoyuki Mizuno

**Affiliations:** 1Division of Translational and Clinical Pharmacology, Cincinnati Children’s Hospital Medical Center, Cincinnati, OH 45229, USA; kei.irie@cchmc.org (K.I.); tomoyuki.mizuno@cchmc.org (T.M.); 2Department of Clinical Pharmacy, Oita University Hospital, Yufu 879-5593, Japan; 3Department of Pediatrics, University of Cincinnati College of Medicine, Cincinnati, OH 45221, USA

**Keywords:** antibiotics, pediatrics, model-informed precision dosing, physiologically based pharmacokinetic modeling

## Abstract

The appropriate use of antibiotics is crucial and involves selecting an optimal dosing regimen based on pharmacokinetic (PK) and pharmacodynamic (PD) indicators. Physiologically based pharmacokinetic (PBPK) modeling is a powerful tool that integrates drugs’ physicochemical properties with anatomical and physiological data to predict PK behavior. In pediatric populations, PBPK modeling accounts for developmental changes in organ function, making it particularly useful for optimizing antibiotic dosing across different age groups, from neonates to adolescents. In recent decades, PBPK modeling has been widely applied to predict antibiotic disposition in pediatric patients for various clinical and research purposes. Model-informed precision dosing (MIPD) is an evolving approach that enhances traditional therapeutic drug monitoring by integrating multiple information sources into a mathematical framework. By incorporating PBPK modeling, MIPD could offer a more optimized antibiotic dosing that accounts for PK/PD parameters at the site of infection, improving therapeutic outcomes while minimizing toxicity. This review summarizes currently published pediatric PBPK modeling studies on antibiotics, covering various objectives such as evaluating drug–drug interactions, PK/PD analyses in targeted tissues, predicting PK in specific populations (e.g., maternal/fetal, renal impairment, obesity), and PK predictions for preterm neonates. Based on these reports, the review discusses the implications of PBPK modeling for MIPD in pediatric antibiotic therapy.

## 1. Introduction

The judicious use of antimicrobials is crucial for minimizing adverse effects, enhancing clinical efficacy, preventing the emergence of drug-resistant bacteria, and achieving economic benefits. Determining an optimal dosing regimen requires consideration of pharmacokinetics (PK)/pharmacodynamic (PD) indices, such as the peak plasma concentration (C_max_)/minimum inhibitory concentration (MIC) ratio, area under the curve (AUC)/MIC ratio, and time above MIC (f_T>MIC_) [1]. However, a one-size-fits-all approach with body weight-based dose adjustments often does not achieve an adequate PK/PD index in individual patients, particularly in pediatric populations. For instance, the variability in vancomycin pharmacokinetics in children is attributed to differences in renal function and maturation, which were not adequately accounted for by body weight-based dosing alone [2]. Optimizing the dosage of antimicrobials in children is crucial for improving treatment outcomes, minimizing toxicity, and combating antimicrobial resistance. Several studies emphasize the importance of therapeutic drug monitoring (TDM) and PK/PD modeling to tailor antimicrobial dosage in children, ensuring effective and safe treatment outcomes [3].

Pediatric dose optimization is inherently more complex than in adults due to the need to consider growth and organ maturation. Simply determining pediatric doses by scaling from adult doses based on body weight may lead to underexposure or overexposure due to their unique physiological characteristics (e.g., non-linear correlation between body weight and clearance and/or effects of maturation), resulting in treatment failure and adverse events [4,5]. While allometric scaling is commonly used to describe the relationship between body weight and drug clearance in children [6,7,8,9], pediatric PK is further influenced by age-related changes in drug absorption, distribution, metabolism, and excretion (ADME). For instance, neonates and infants may experience reduced expression and/or activity of drug-metabolizing enzymes and transporters, with various maturation rates across different organs [10,11]. These developmental differences can significantly impact drug exposure and toxicity risks [12], making precise dose adjustments crucial. Furthermore, to ensure precise dose adjustment based on PK/PD prediction, developing drug dosage forms that can be administered flexibly to children, such as powder or liquid formulations, is essential. This necessitates a multifaceted approach that addresses manufacturing considerations, such as taste, swallowability, and the use of safe excipients, and ensures dose uniformity.

Physiologically based PK (PBPK) modeling is a mechanistic approach to addressing these challenges by integrating drug-specific properties with age-dependent physiological changes. Unlike empirical scaling methods, PBPK models incorporate detailed anatomical and physiological data to simulate drug movement through various organ compartments, capturing developmental changes in distribution and clearance mechanisms [13]. This mechanistic approach allows for improved predictions of drug behavior across pediatric subpopulations, from neonates to adolescents [14,15]. PBPK models have been widely applied to predict antibiotic disposition in specific populations, including preterm neonates, infants with renal impairment, and children with obesity [16]. These models have also been used to assess drug–drug interactions (DDIs), optimize age-specific dosing strategies, and predict antibiotic penetration into target tissues.

Model-informed precision dosing (MIPD) is an emerging framework that integrates multiple data sources to facilitate personalized dose optimizations, and it has been applied to maximize the success of antimicrobial therapy [17]. The MIPD approach has also been implemented to optimize pediatric pharmacotherapy, where the age-dependent variability in drug metabolism and clearance is particularly pronounced. Given the critical importance of PK/PD target attainment in antimicrobial therapy, PBPK-based MIPD approaches hold significant potential for optimizing antibiotic use in children, minimizing toxicity, and improving treatment outcomes.

This review provides an overview of the pediatric PBPK modeling studies of antibiotics published over the last decade. Furthermore, their implications and prospects for MIPD in pediatric antibiotic therapy are discussed based on these reports.

## 2. Pediatric PBPK Modeling

### 2.1. Pediatric PBPK Modeling Overview

Several review articles have comprehensively summarized adult and pediatric PBPK modeling [16,18,19,20,21]. This review, therefore, focuses on a summary of published pediatric antibiotic PBPK modeling and briefly describes the pediatric PBPK modeling overview. Pediatric PBPK modeling is a mechanistic approach used to predict drug ADME in children by incorporating age-specific physiological, biochemical, and molecular data [22]. Key physiological differences in children, such as body composition, enzyme maturation, renal function, plasma protein binding, and gastrointestinal development, significantly impact drug pharmacokinetics [23]. In pediatric patients, body composition is distinct, with infants exhibiting approximately 22.4% fat at 12 months, compared to 13% in older children, and extracellular water comprising up to 70% of total body weight in neonates. Plasma protein binding is reduced, with newborns displaying protein levels at roughly 86% of adult values, increasing free drug concentrations. Hepatic enzyme maturation shows microsomal protein content at 26 mg/g in neonates, rising to 40 mg/g in adults, influencing metabolic clearance. Renal function is immature with a GFR of 2–4 mL/min/1.73 m^2^ in term neonates, achieving adult levels by the end of the first year. Data sources include clinical pediatric studies, in vitro enzyme activity, anatomical databases, and adult PBPK models extrapolated to the pediatric context. These models are widely used for dose selection, DDI prediction, extrapolation from adults to children, and regulatory submissions; however, challenges such as limited pediatric-specific data, transporter maturation uncertainty, and population variability exist. Regulatory agencies, including the U.S. Food and Drug Administration (FDA) and the European Medicines Agency, support PBPK modeling for pediatric drug development [24], and future advancements aim to improve physiological databases and transporter modeling for enhanced predictions. Pediatric PBPK modeling continues to evolve as a vital tool for optimizing pediatric drug therapy while reducing clinical study burdens.

### 2.2. Applications of Pediatric PBPK Modeling

Pediatric PBPK modeling has been widely applied across various therapeutic areas to enhance drug dosing precision and safety [16]. It plays a critical role in predicting DDIs by accounting for age-dependent enzyme maturation and transporter activity, which can significantly alter drug metabolism in children compared to adults [25]. It is also used to estimate drug concentrations in specific tissues, such as the lungs, cerebrospinal fluid, and heart, improving the understanding of drug distribution at infection or target sites [26]. In maternal–fetal applications, PBPK models help predict fetal drug exposure, optimizing maternal dosing while ensuring effective and safe drug levels in the fetus [27,28]. For preterm neonates, PBPK modeling addresses challenges in dose selection by incorporating developmental changes in organ function, metabolism, and clearance [29]. In further specific populations, such as children with obesity or renal impairment, these models can adjust for physiological alterations that affect drug distribution and elimination, providing more accurate dosing recommendations [30,31].

## 3. Published Pediatric PBPK Models of Antibiotics

### 3.1. Literature Search

For this narrative review, we conducted a systematic literature search partially following the Preferred Reporting Items for Systematic Reviews and Meta-Analyses guidelines (Appendix A) [32]. A comprehensive search of PubMed (National Center for Biotechnology Information, National Institutes of Health) was performed between 1 January 2016 and 22 November 2024, using the following search terms: ((Physiologically based pharmacokinetic) OR (PBPK)) AND ((Pediatric) OR (Paediatric) OR (Children) OR (Adolescent) OR (Infant) OR (Newborn) OR (Neonate) OR (Preterm)) AND antibiotic. The final data set (Articles initially retrieved.xlsx) in an Excel format is available in Appendix A.

### 3.2. Included Articles

A total of 89 articles were identified, with no duplicates (Figure 1). Articles were excluded if they did not address PBPK modeling, were unrelated to antibiotics or pediatrics, or were reviews. In the end, 27 articles met the inclusion criteria (Appendix A). Each study’s objective was classified into one of six categories: (1) dose selection for children of different ages; (2) DDI evaluation; (3) PK/PD analysis in targeted tissues; (4) PK prediction in specific pediatric populations, such as maternal/fetal, renal impairment, or obesity; (5) PK prediction for preterm neonates; or (6) other (Figure 2). Out of the 27 articles, those that are particularly aligned with each category are described in this manuscript.

### 3.3. Dose Selection for Children of Different Ages

Five pediatric antibiotic PBPK models have aimed to determine the appropriate dosing regimen across diverse pediatric ages [33,34,35,36,37]. Hornik et al. [33] developed a pediatric PBPK model for clindamycin, a commonly used antibiotic. The PBPK model incorporated age-dependent physiological parameters, including glomerular filtration rate (GFR) and tubular secretion via OAT-1, to simulate drug disposition in children. The model proposed the optimal dosing regimens to achieve the PK/PD indices in four different pediatric age groups. The model’s predictions aligned with existing dosing guidelines for methicillin-resistant *Staphylococcus aureus* (MRSA) treatment, supporting its clinical applicability. Thompson et al. [34] developed a pediatric PBPK model for trimethoprim (TMP) and sulfamethoxazole (SMX) to optimize dosing in five different age groups of children, particularly for treating MRSA infections. The dosing simulations demonstrated that the optimized pediatric dosing regimens achieved adequate skin concentrations, surpassing MIC thresholds toward MRSA for the majority of patients. The model-informed optimal dosing of TMP was in the range of recommended dosing for TMP-SMX in children of all ages, as outlined in current guidelines for MRSA treatment. Liang et al. [37] developed a PBPK model to extrapolate adult azithromycin data to pediatrics to optimize pediatric dosing for different age groups. The model incorporated age-dependent algorithms to adjust parameters such as plasma protein binding, renal clearance, and gastric pH for pediatric patients. The final model provides dosing recommendations for different pediatric age groups in treating community-acquired pneumonia. Table 1 presents the optimized dosing regimens for antibiotics in children of different ages, as proposed by the PBPK model used in each study. Among these drugs, the dosing regimens for only moxifloxacin were prospectively validated in phase I and III clinical trials. The PBPK model-informed daily doses tend to be highest in children aged 2–6 years and decrease as the child grows. This is partially attributed to the maturation of transporter and metabolic enzyme activities around two years of age, as well as the allometric relationship between organ clearance and body size.

### 3.4. DDI Evaluation

The FDA strongly recommends clinical DDI studies for the metabolism and inhibition of drug-metabolizing enzymes, especially CYP3A [38]. Clinical DDI studies are typically conducted in adults; however, they are not routinely performed in pediatric patients due to ethical and practical challenges. However, given that maturation affects drug-metabolizing enzymes and transporters, the extent of DDIs in pediatric patients may differ substantially from adults. Indeed, a systematic review comparing pediatric and adult studies revealed that developmental changes in drug-metabolizing enzymes and transporters can lead to significant differences in the magnitude of DDIs [39]. Salerno et al. [40] developed a framework using PBPK modeling to forecast the potential DDI in pediatric patients, using solithromycin, a substrate and time-dependent inhibitor of CYP3A4. The PBPK models incorporated in vitro data for CYP3A4, CYP3A5, and CYP3A7 to predict solithromycin’s PK in adult and pediatric populations. Furthermore, the DDI of solithromycin with midazolam and ketoconazole was evaluated across different age groups. The authors demonstrated that solithromycin increased the simulated midazolam AUC by four- to six-fold, while ketoconazole increased the simulated solithromycin AUC by one- to two-fold, with differences observed among age groups. This research offered a systematic approach for integrating CYP3A in vitro data into PBPK models, enhancing the prediction of pediatric CYP3A-mediated DDI. Litjens et al. [41] developed a PBPK model to understand the interactions between moxifloxacin and rifampicin and to optimize moxifloxacin dosing in the treatment of tuberculosis. Moxifloxacin is metabolized primarily by UGT1A1 and SULT2A1, with P-glycoprotein influencing absorption and excretion [42,43,44]. Rifampicin induces these enzymes [45,46], leading to reduced moxifloxacin plasma concentrations and potential resistance. Their results indicated that dose adjustment can restore sufficient moxifloxacin plasma exposure in pediatric patients with tuberculosis when co-administered with rifampicin.

### 3.5. PK/PD Analyses in Targeted Tissues

Drug concentration at the infection site is a key determinant of antibiotic efficacy [47,48]. Plasma concentrations often serve as proxies for tissue drug levels, but actual tissue penetration can vary [49]. To date, four pediatric antibiotic PBPK models have been developed to analyze the PK/PD profiles in specific tissues [50,51,52,53]. For instance, Zhu et al. [51] used PBPK models to predict the disposition and efficacy of polymyxins, meropenem, and sulbactam against multidrug-resistant *Acinetobacter baumannii* in diverse pediatric tissues, such as the blood, skin, lungs, and heart, by linking drug exposures to PD indices. The recommended pediatric dosing regimens provided sufficient drug exposure in targeted tissues, supporting their use in treating infections in children. The same group also employed PBPK modeling of polymyxin-B, amikacin, and sulbactam to predict tissue exposures in lung, skin, and heart tissues, to evaluate the efficacy of these drugs against multidrug-resistant *Acinetobacter baumannii* [53]. Their previous study showed that this triple combination narrowed the mutant selection window and reduced the likelihood of multidrug-resistant *Acinetobacter baumannii* acquiring further resistance [54]. Their simulations indicated that the PK/PD breakpoints were achieved in 85–90% of the virtual pediatric population aged 2 to <18 years. This suggests that the combination therapy of amikacin, polymyxin-B, and sulbactam can effectively penetrate tissues to combat extremely drug-resistant *Acinetobacter baumannii* clinical isolates. It should be noted, however, that as these two established PBPK models were not verified using observed tissue concentrations, the accuracy of the predicted tissue exposures remains unclear. Minimally or non-invasive approaches, such as microdialysis and imaging techniques, may have the potential to bridge this gap.

Linezolid has potential efficacy against multidrug-resistant tuberculosis [55], but its PK profile and target attainment in cranial cerebrospinal fluid (CSF) remains unknown. To address this knowledge gap, a linezolid PBPK model has been developed to forecast the CSF levels in adult and pediatric patients with tuberculous meningitis [52]. The model was refined using data from the literature and experiments on linezolid transport across the blood–brain barrier, employing a four-compartment CSF model using the Simcyp simulator [56]. The model-based simulation indicates that a daily 1200 mg dose in adults may achieve reasonable target attainment in cranial CSF, while pediatric target attainment (10 mg/kg twice a day) was moderate. Furthermore, the study concluded that this model can be used to estimate the CSF disposition of linezolid in cases where plasma concentration measurements for individual patients are available. Table 2 indicates the PBPK model-informed PK/PD breakpoints in specific tissues or percentages reaching the PK/PD threshold for each antibiotic. These values vary significantly across age groups, antibiotics, and tissues; therefore, PBPK modeling may be an optimal precision dosing strategy for each target tissue in children.

### 3.6. PK Prediction in Specific Pediatric Populations

Predicting PK in children with renal impairment, obesity, or in fetuses is often challenging due to limited clinical data. PBPK modeling offers a promising alternative approach for predicting drug exposure in these populations. To date, seven pediatric antibiotic PBPK models have been developed to predict PK in specific pediatric populations, including those with renal impairment [57,58,59,60,61], obesity [62], and in fetuses [63]. Four PBPK models, of ertapenem [58], ceftazidime [59], ceftaroline [60], and teicoplanin [61], were developed by the same research groups to identify the most appropriate dosing regimens of each antibiotic for pediatric patients with estimated GFRs of 60–89, 30–59, 15–29, and <15 mL/min/1.73 m^2^. The dosages of these antibiotics may be tailored in accordance with the patient’s renal function; however, the optimal dosing regimens for pediatric patients with renal impairment have yet to be determined. The developed PBPK models adequately predicted drug exposures in patients with varying degrees of renal impairment and provided optimal dosing regimens to maintain therapeutic efficacy for these populations (Table 3).

The growing prevalence of childhood obesity represents a significant public health concern, with rates nearly quadrupling over the past 25 years [64]. Obesity significantly influences drugs’ PK due to changes in body size and composition, which affect drug distribution and elimination [65,66]. To address the lack of dosing guidance for children with obesity, Gerhart et al. [62] devised a virtual population of children with obesity to facilitate PBPK modeling. This model incorporated obesity-related physiological changes, such as increased body size, organ size, and altered blood flow, and was used to predict the PK profiles of clindamycin and trimethoprim/sulfamethoxazole. It indicated that current weight-based dosing recommendations were adequate, despite drug clearance and distribution alterations.

Ampicillin is commonly used in neonates to treat sepsis and as an intrapartum prophylaxis for Group B *Streptococcus* [67,68]. However, clinical PK studies in these vulnerable populations, particularly in fetuses, are constrained by various challenges. Fetomaternal PBPK modeling is an emerging tool for predicting drug PK in pregnant women and fetuses [27,28,69]. Li et al. [63] applied such models to optimize ampicillin dosing for these populations. The model was initially developed for non-pregnant adults and subsequently adapted for neonates and pregnant women to account for physiological alterations and variations in drug disposition. The study demonstrated that a 50 mg/kg every 8 h regimen achieved desired PD targets in neonates, while a 1 g maternal dose provided adequate fetal exposure for 4 h before delivery.

### 3.7. PK Prediction for Preterm Neonates

Various constraints and recruitment challenges make it difficult to conduct robust clinical trials in preterm neonates. Consequently, dose selection often relies on empirical methods, overlooking developmental differences. Neeli et al. [70] constructed a gentamicin PBPK/PD model to inform dosing decisions in preterm neonates and to predict the antibacterial effects. The PBPK model was initially verified in healthy adults and subsequently extended to term and preterm neonates across various gestational ages (GAs). The simulation indicated that a higher dose with an extended dosing interval (every 36 h) is more effective for maintaining safe trough levels compared to once-daily dosing, the standard dosing regimen obtained from Neofax [71], particularly in neonates with a postmenstrual age (PMA) of 30–34 weeks with a postnatal age (PNA) of 8–28 days and with a PMA of ≥35 weeks with a PNA of 0–7 days. Ganguly et al. [72] developed a PBPK model for meropenem, a broad-spectrum carbapenem antibiotic, in preterm and term infants to gain insight into the impact of maturation in GFR and transporter-mediated tubular secretion. Their model adequately predicted plasma concentrations in preterm and term infants and successfully captured the post hoc estimated clearance of meropenem. While GA was found to influence meropenem clearance significantly, the dosing regimens recommended in the product labeling for infants under three months of age were deemed sufficient for achieving the PK/PD target. Li et al. [73] developed a PBPK model for cefotaxime, a commonly used antibiotic in neonates, to optimize dosing regimens for preterm and term neonates. The model initially developed for adults incorporated three elimination pathways: (1) enzymatic metabolism in the liver, (2) passive glomerular filtration, and (3) active renal tubular secretion mediated by organic anion transporters. The model was then scaled to neonates by accounting for age-dependent physiological changes, such as GFR and organ maturation. The study underscored the significant effect of GA and PNA on cefotaxime PK in neonates. Based on these parameters, the following dosing regimens were proposed: 25 mg/kg every 8 h in PNA between 0 and 6 days and every 6 h in PNA between 7 and 28 days for preterm neonates (GA < 36 weeks); 33 mg/kg every 8 h in PNA between 0 and 6 days and every 6 h in PNA between 7 and 28 days for term neonates (GA ≥ 36 weeks). Table 4 provides optimal dosing scenarios in preterm neonates across various GAs, as predicted by each PBPK model.

### 3.8. Other Studies

Six additional studies developed pediatric PBPK models of antibiotics with diverse aims [74,75,76,77,78,79]. Duan et al. [74] developed PBPK models for linezolid and emtricitabine in neonates and infants, emphasizing the importance of accounting for renal maturation. Nguyen et al. [75] highlighted how flavin-containing monooxygenase 3 (FMO3) ontogeny affects ethionamide metabolism, suggesting that PBPK models can help refine pediatric dosing to improve efficacy and reduce toxicity. Martins et al. [76] evaluated the co-administration of meropenem and fosfomycin in pediatric patients with multidrug-resistant bacteria such as *Klebsiella pneumoniae* and *Pseudomonas aeruginosa*. Their study identified effective dosing regimens, such as 20 mg/kg meropenem and 35 mg/kg fosfomycin every 8 h as 3 h infusions, to achieve the desired PTA in a virtual pediatric population. Idkaidek et al. [77] developed a gentamicin PBPK model to facilitate TDM using saliva samples in Jordanian preterm infants. The model successfully predicted that gentamicin concentrations in saliva are 57 times higher than in plasma, aligning with observed results. This suggests that saliva may be a viable alternative for TDM in preterm infants. Guimarães et al. [78] employed in vitro tests and PBPK modeling to elucidate the influence of solubility, dissolution, and permeability on the absorption of azithromycin and to evaluate the biopharmaceutical risk associated with age-related alterations in its oral performance. The findings indicated that permeability is critical for influencing the absorption of azithromycin, with passive and active transport processes serving as rate-limiting steps.

## 4. Future Perspectives and Current Challenges for PBPK-Facilitated MIPD in Pediatric Antibiotic Therapy

Besides empirical, clinical data-driven (top-down) population PK modeling, PBPK approaches have been gaining attention for MIPD in pediatric antibiotics therapy to account for mechanistic insights into drug disposition and complement clinical data gaps, especially for specific populations where clinical data are scarce [17]. While most antimicrobial dosing guidelines for children are outlined in package inserts and various recommendations, they typically do not address specific pediatric subpopulations, such as obese children, those with renal impairment, or premature neonates. Instead, dosing for these groups is often extrapolated from regimens designed for typical pediatric patients, potentially leading to both overexposure and underexposure [80]. This imbalance in drug exposure increases the risks of therapeutic failure, antimicrobial resistance, and adverse events. The rarity of these specific subpopulations poses a challenge for generating robust population PK models due to insufficient clinical data [81]. In contrast, PBPK models are built on a mechanistic understanding of biological processes, leveraging physiological and anatomical data to predict PK even when clinical data are sparse. This mechanistic basis approach could provide more robust PK prediction and rational dosing strategies for vulnerable pediatric groups.

A key advantage of PBPK modeling lies in its capacity to estimate tissue-specific drug concentrations based on the physiological mechanisms of drug distribution. Since most antimicrobials exert their primary effects at the site of infection (with the exception of bacteremia and catheter-related infections), tissue exposure is critical for predicting therapeutic efficacy. Nevertheless, most antimicrobial regimens are determined based on PK/PD parameters derived from plasma concentrations. These regimens are generally applied irrespective of the infected tissue, except for meningitis. PBPK model-predicted tissue drug concentrations could offer valuable insights into characterizing PK/PD relationship at the site of action (infection). A primary hurdle in leveraging PBPK models for tissue-specific drug exposure predictions is validating these estimates, given the challenges of directly measuring drug concentrations in human tissues. To address this, non-invasive imaging techniques, such as positron emission tomography and magnetic resonance imaging, are increasingly used to corroborate PBPK model-based predictions of drug tissue distribution [82,83]. Early findings suggest that these imaging-based validation approaches can bolster confidence in PBPK estimates of drug concentrations in human tissues.

In addition to the above modeling efforts, there has been a growing body of research aimed at developing digital twins, computational representations of individual patients, to predict drug responses closely reflecting real-world clinical scenarios. PBPK model-based “virtual twins” are increasingly used to simulate drug dispositions in a specific patient by incorporating factors such as age, body weight, sex, renal and hepatic function, genetic factors, and comorbid conditions [84]. Now, researchers are integrating actual patients with pediatric virtual twins in clinical settings, where real-world data from patients are integrated into PBPK models to refine treatments with various drugs such as olanzapine, caffeine, enoxaparin, and clozapine [85,86,87,88]. Moreover, the addition of tissue-specific PK/PD considerations to virtual twin frameworks may facilitate individualized dosing based on localized drug exposure predictions, which is especially vital for infections or targeted therapies. However, broader implementation requires addressing two main challenges [84]. First, disease-specific pathophysiology should be adequately captured in the models, including potential interactions of disease severity/improvement and ADME processes. Second, constructing a high-fidelity virtual twin demands diverse and reliable patient data, such as genetic profiles and environmental factors. Indeed, previous studies using the virtual twins approach demonstrated an overprediction of steady-state clozapine concentrations in patients concomitantly treated with fluvoxamine, highlighting the need for model refinement to account for DDIs [88]. Future research could enhance the granularity of input data and improve modeling algorithms to make digital twins a cornerstone of personalized medicine.

In recent years, virtual imaging trials and digital twins have emerged as an innovative approach in medical research, addressing the limitations of traditional clinical trials, such as high costs and ethical constraints [89]. Regulatory authorities, including the FDA, underscore the imperative for robust simulation validation, transparency in methodological execution, and adherence to Good Simulation Practices to ensure trial credibility and reproducibility. While they do not provide their views on applying imaging techniques or virtual twins in PBPK modeling, they acknowledge the potential of these innovative approaches in future clinical trials and practices. However, the FDA maintains a cautious stance, underscoring the necessity of rigorous validation and regulatory evidence.

PBPK modeling has also gained significant traction for formulation optimization via in vitro–in vivo correlations [90]. Although detailed exploration of this application falls outside the scope of this review, it is noteworthy that PBPK could greatly facilitate the design of pediatric-friendly dosage forms. Such optimization enables flexible dose adjustments, enhancing both therapeutic effectiveness and patient compliance in children.

Finally, while the use of PBPK in antimicrobial MIPD shows promise, several limitations remain. First, PBPK modeling depends on the reliability of physiological and pharmacological assumptions and their mathematical representation. Each simulation using a developed PBPK model represents a simplified version of complex real-world processes and is inherently subject to error. These errors can accumulate and amplify in the model output, resulting in biased predictions. Second, pediatric PBPK modeling is constrained by the limited and heterogeneous clinical pharmacokinetic data available, particularly in the cases of neonates and preterm infants [91]. This scarcity of empirical data hinders the validation of established models, leading to reliance on extrapolated or assumed data derived from adult physiology. This, in turn, amplifies uncertainty in predictions. In addition, the limited availability of robust ontogeny data for drug-metabolizing enzymes and transporters further complicates the accurate depiction of maturation processes. Third, the translation of PBPK models into clinical practice remains limited, compared to the use of top-down population PK/PD modeling approaches. Developing user-friendly platforms that seamlessly integrate PBPK modeling into clinical workflows will be pivotal for broader adoption. In parallel, clinical evidence is needed to substantiate the real-world benefits of these platforms, ultimately driving the evolution of precision antibiotic dosing in pediatric populations.

## 5. Conclusions

PBPK modeling has emerged as a powerful tool for optimizing pediatric antibiotic therapy by incorporating physiological, biochemical, and developmental factors into drug disposition predictions. Over the past decade, numerous pediatric PBPK models have been published for antibiotics, covering a wide range of applications, including dose selection across different age groups; DDI evaluation; PK/PD analyses in targeted tissues; and the prediction of drug behavior in specific pediatric populations, such as those with renal impairment, obesity, or preterm neonates. These models provide critical insights that support safer and more effective antibiotic dosing in children, particularly in the context of MIPD. Despite its potential, the clinical application of PBPK-based MIPD remains limited, necessitating further validation studies and real-world implementation. Integrating PBPK-based virtual twins approaches into MIPD presents a promising strategy for dose individualization, yet challenges remain in refining patient-specific characteristics and incorporating disease-related variations into the models. Continued collaboration among researchers, clinicians, and regulatory agencies will be essential for facilitating the broader clinical adoption of PBPK-based MIPD and, ultimately, improving individualized antibiotic therapy in pediatric patients.

## Figures and Tables

**Figure 1 antibiotics-14-00541-f001:**
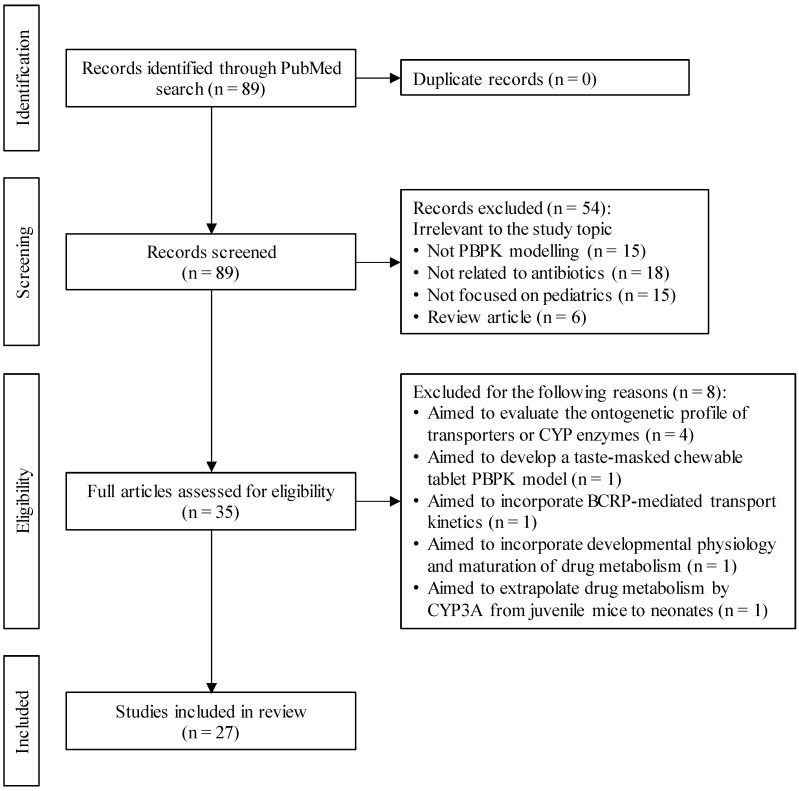
Flowchart for the identification of previous pediatric physiologically based pharmacokinetic modeling studies for antibiotics.

**Figure 2 antibiotics-14-00541-f002:**
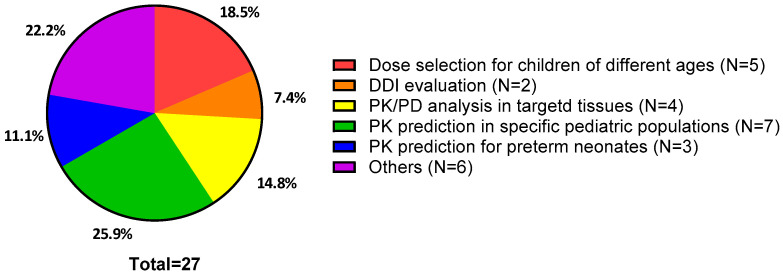
The objectives of pediatric physiologically based pharmacokinetic modeling studies. DDI: drug–drug interaction; PK: pharmacokinetics; PD: pharmacodynamics.

**Table 1 antibiotics-14-00541-t001:** Optimized dosing regimens for each antibiotic in children of different ages using developed PBPK models.

Study Drug	Age	Optimized Dosing Regimen	Reference	Current Pediatric Dosing Regimen	Adult Dosing Regimen
Clindamycin	0–5 months	9 mg/kg every 8 h *^1^	Hornik CP et al., *Clin Pharmacokinet*. 2017 [33]	20–40 mg/kg/day divided into 6–8 h	600 mg every 8 or 12 h
	>5 months–1 year	12 mg/kg every 8 h *^1^			
	>1–6 years				
	>6–12 years	10 mg/kg every 8 h *^1^			
	>12–18 years				
Trimethoprim (sulfamethoxazole)	>2–5 months	30 mg/kg (6 mg/kg) every 12 h *^2^	Thompson EJ et al., *Clin Pharmacokinet*. 2019 [34]	8–12 mg/kg/day (40–60 mg/kg/day) divided into 12 h	160 mg (800 mg) every 12 h
	>5 months–1 year				
	>1–6 years				
	>6–12 years				
	>12–18 years	20 mg/kg (4 mg/kg) every 12 h *^2^			
Moxifloxacin	3 months–<2 years	9–10 mg/kg every 12 h	Willmann S et al., *CPT Pharmacometrics Syst Pharmacol*. 2019 [35]	7.5–10 mg/kg every 24 h	400 mg every 24 h
	2–<6 years	7–8 mg/kg every 12 h			
	6–<12 years	5–6 mg/kg every 12 h			
	12–<18 years	200 mg every 12 h (>45 kg)			
Daptomycin Ceftaroline	2–<6 years	7 mg/kg every 24 h *^3^ 12 mg/kg every 8 h *^3^	Martins FS et al., *Br J Clin Pharmacol*. 2023 [36]	5–9 mg/kg every 24 h 12 mg/kg every 8 h	4–6 mg/kg every 24 h 600 mg every 24 h
	6–<12 years				
	12–<18 years				
Azithromycin	0.5–2 years	Day 1: 8.8 mg/kg; Days 2–5: 4.4 mg/kg *^4^	Liang L et al., *Biopharm Drug Dispos*. 2023 [37]	10 mg/kg every 24 h	500 mg every 24 h
	3–6 years	Day 1: 9.2 mg/kg; Days 2–5: 4.6 mg/kg *^4^			
	7–12 years	Day 1: 9.4 mg/kg; Days 2–5: 4.7 mg/kg *^4^			
	13–18 years	Day 1: 8.2 mg/kg; Days 2–5: 4.1 mg/kg *^4^			

*^1^ MRSA with MIC of 0.5 μg/mL or below; *^2^ MRSA with MIC of 2 μg/mL/9.5 μg/mL or below; *^3^ MRSA with MIC of 0.5–2 μg/mL; *^4^ the maximum first dose was capped at 500 mg, and the daily dose from Days 2 to 5 should not exceed 250 mg. PBPK: physiologically based pharmacokinetics.

**Table 2 antibiotics-14-00541-t002:** PBPK model-informed PK/PD breakpoint or percentage reaching threshold of PK/PD indices in focused tissues in each study.

Study Drug	Tissue	Age (Years)	Dose	AUC Ratio (Plasma/Tissue)	PK/PD Indices	PK/PD Breakpoint or Reaching Threshold	Reference
Colistin	Plasma	2–<6	5 mg/kg q8h	1	fAUC/MIC ≥ 7.4	1 µg/mL	Zhu S et al., *Clin Pharmacokinet*. 2022 [51]
		6–<12		1		2 µg/mL	
		12–<18		1		2 µg/mL	
	Heart	2–<6		0.15		0.5 µg/mL	
		6–<12		0.15		1 µg/mL	
		12–<18		0.15		1 µg/mL	
	Lung	2–<6		0.56		1 µg/mL	
		6–<12		0.56		1 µg/mL	
		12–<18		0.56		1 µg/mL	
	Skin	2–<6		0.42		2 µg/mL	
		6–<12		0.42		2 µg/mL	
		12–<18		0.42		4 µg/mL	
Meropenem	Plasma	2–<6	30 mg/kg q8h	1	fT > MIC ≥ 40%	8 µg/mL	
		6–<12		1		8 µg/mL	
		12–<18		1		8 µg/mL	
	Heart	2–<6		0.21		1 µg/mL	
		6–<12		0.2		2 µg/mL	
		12–<18		0.2		2 µg/mL	
	Lung	2–<6		0.37		2 µg/mL	
		6–<12		0.37		2 µg/mL	
		12–<18		0.37		4 µg/mL	
	Skin	2–<6		0.63		4 µg/mL	
		6–<12		0.63		4 µg/mL	
		12–<18		0.63		8 µg/mL	
Sulbactam	Plasma	2–<6	40 mg/kg q8h	1	fT > MIC ≥ 60%	4 µg/mL	
		6–<12		1		4 µg/mL	
		12–<18		1		8 µg/mL	
	Heart	2–<6		0.51		2 µg/mL	
		6–<12		0.51		2 µg/mL	
		12–<18		0.51		4 µg/mL	
	Lung	2–<6		0.51		4 µg/mL	
		6–<12		0.52		4 µg/mL	
		12–<18		0.5		8 µg/mL	
	Skin	2–<6		0.28		2 µg/mL	
		6–<12		0.29		2 µg/mL	
		12–<18		0.27		2 µg/mL	
Polymyxin-B	Plasma	2–<18	1.25 mg/kg q12h	1	fAUC_0–24_/MIC ≥ 8.2	2 µg/mL	Wu M et al., *Front Microbiol*. 2024 [53]
	Heart			1.07		4 µg/mL	
	Lung			2.99		4 µg/mL	
	Skin			1.56		8 µg/mL	
Amikacin	Plasma		15 mg/kg q12h	1	fAUC_0–24_/MIC ≥ 80	4 µg/mL *^1^	
	Heart			1.1		4 µg/mL *^1^	
	Lung			0.51		1 µg/mL *^1^	
	Skin			0.42		1 µg/mL *^1^	
Sulbactam	Plasma		1.5 g q6h	1	fT > MIC ≥ 40%	8 µg/mL *^1^	
	Heart			0.51		4 µg/mL *^1^	
	Lung			0.51		4 µg/mL *^1^	
	Skin			0.29–0.31		4 µg/mL *^1^	
Linezolid	Plasma	0.25–<21	10 mg/kg q12h	1	AUC/MIC > 119	93% *^2^	Litjens CHC et al., *Antibiotics (Basel)*. 2023 [52]
					T > MIC > 80%	100% *^2^	
	Cranial CSF			0.67	AUC/MIC > 119	56% *^2^	
					T > MIC > 80%	100% *^2^	

*^1^ Children weighing ≥ 40 kg; *^2^ Percentage reaching AUC/MIC threshold of 119 with MIC 1 mg/L. PK: pharmacokinetics; PD: pharmacodynamics; AUC: area under the curve; MIC: minimum inhibitory concentration; CSF: cerebrospinal fluid. PBPK: physiologically based pharmacokinetics.

**Table 3 antibiotics-14-00541-t003:** PBPK model-informed optimal antibiotic dosing scenarios in pediatric patients with renal impairment.

Study Drug	Age	eGFR (mL/min/1.73 m^2^)	Optimized Dosing Regimen	Reference
Ertapenem	≤12 years	60–89	13 mg/kg every 2 h *^2^	Ye L et al., *J Pharm Sci*. 2020 [58]
		30–59	9 mg/kg every 2 h *^2^	
		15–29	6 mg/kg every 2 h *^2^	
		<15	5 mg/kg every 2 h *^2^	
Ceftazidime	1 month–12 years *^1^	60–89	50 mg/kg every 8 h *^2^	Zhou J et al., *J Pharm Sci*. 2021 [59]
		30–59	28 mg/kg every 8 h *^2^	
		15–29	15 mg/kg every 8 h *^2^	
		<15	8 mg/kg every 8 h *^2^	
Ceftaroline	2–<18 years	60–89	12 mg/kg every 8 h *^2^	Zhou J et al., *J Clin Pharmacol*. 2021 [60]
		30–59	8 mg/kg every 8 h *^2^	
		15–29	6 mg/kg every 8 h *^2^	
		<15	5 mg/kg every 8 h *^2^	
	2 months–<2 years	60–89	8 mg/kg every 8 h *^2^	
		30–59	5 mg/kg every 8 h *^2^	
		15–29	4 mg/kg every 8 h *^2^	
		<15	3 mg/kg every 8 h *^2^	
	0–<2 months	60–89	6 mg/kg every 8 h *^2^	
		30–59	4 mg/kg every 8 h *^2^	
		15–29	3.5 mg/kg every 8 h *^2^	
		<15	2.5 mg/kg every 8 h *^2^	
Teicoplanin	2–12 years	60–89	12 mg/kg every 12 h *^3^	Xu J et al., *J Clin Pharmacol*. 2022 [61]
		30–59	9.5 mg/kg every 12 h *^3^	
		15–29	6 mg/kg every 12 h *^3^	
		<15	4 mg/kg every 12 h *^3^	

*^1^ Same suggested adult dosing regimen for children 12 years of age and older weighing more than 40 kg; *^2^ MIC of 4 mg/L or below; *^3^ loading dose for MRSA with MIC of 1 mg/L or below. eGFR: estimated glomerular filtration rate; MRSA: methicillin-resistant *Staphylococcus aureus*; MIC: minimum inhibitory concentration.

**Table 4 antibiotics-14-00541-t004:** PBPK model-informed optimal antibiotic dosing scenarios in preterm neonates across various GAs.

Study Drug	GA or PMA	PNA	Optimized Dosing Regimen	Reference
Gentamicin	PMA 30–34 weeks	0–7 days	4.5 mg/kg every 36 h *^1^	Neeli H et al., *J Clin Pharmacol*. 2021 [70]
		8–28 days	5 mg/kg every 36 h	
	PMA ≥ 35 weeks	0–7 days	5 mg/kg every 36 h	
		8–28 days	4 mg/kg every 24 h *^1^	
Meropenem	GA < 32 weeks	<14 days	20 mg/kg every 12 h	Ganguly S et al., *Clin Pharmacokinet*. 2021 [72]
		>=14 days	20 mg/kg every 8 h	
	GA ≥ 32 weeks	<14 days	20 mg/kg every 8 h	
		>=14 days	30 mg/kg every 8 h	
Cefotaxime	GA < 36 weeks	0–6 days *^2^	25 mg/kg every 8 h	Li Q et al., *J Pharm Sci*. 2024 [73]
		7–28 days *^3^	25 mg/kg every 6 h	
	GA ≥ 36 weeks	0–6 days *^2^	33 mg/kg every 8 h	
		7–28 days *^3^	33 mg/kg every 6 h	

*^1^ Standard dosing regimen from Neofax (the study did not provide the suggested regimen); *^2^ assumed Enterobacteriaceae; *^3^ assumed *Staphylococcus aureus*. GA: gestational age; PMA: postmenstrual age; PNA: postnatal age.

## Data Availability

Not applicable. Only previously published data were used in this manuscript.

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
