# Peer review of "Physiologically Based Pharmacokinetic Modeling of Antibiotics in Children: Perspectives on Model-Informed Precision Dosing"

_antibiotics, 2025, doi:10.3390/antibiotics14060541_

Round 1
Reviewer 1 Report
Comments and Suggestions for Authors
This review article, titled “Physiologically based pharmacokinetic modeling of antibiotics in children: Perspectives on model-informed precision dosing,” gives a clear and thorough overview of how PBPK modeling is used to improve antibiotic dosing in children. It shows how this method helps adjust doses for different age groups and special cases like kidney problems, obesity, and premature babies. The main strengths of the paper are its well-organized review of 27 studies, clear explanation of what is still unknown, and discussion of future tools like virtual twins. Overall, this is a useful and timely contribution to the field of pediatric medicine and personalized antibiotic dosing.
General comment:
This review does a great job of covering all the important areas related to PBPK modeling for antibiotics in children. It talks about how to choose the right dose, how drugs interact with each other, how drugs work in different parts of the body, and how this modeling can help in special groups like newborns, children with obesity, or kidney problems. The authors also point out what’s missing, especially the need to confirm how well these models predict drug levels in tissues and how to use them more in real-world practice. They explain both the potential and the limits of using “virtual twin” models to personalize treatment. The references are appropriate, mostly recent, and come from good-quality sources, with no overuse of self-citations. Overall, this review is unique and timely, especially since no recent papers have focused so specifically on this topic in children.
Specific issues to address:Line 113–120 (Literature Search):
The systematic search is well described. It would enhance transparency if the PRISMA checklist or a brief mention of any protocol registration (e.g., PROSPERO) was included, even if not mandatory for narrative reviews.
Figures 1 and 2:
Figures effectively illustrate the flow of article selection and categorization of study objectives. Consider adding percentages or exact numbers on the bars in Figure 2 for clarity.
Table 1 (Line 165 onward):
The dosing regimens provided are valuable for clinicians. It would be helpful to clarify whether any of these regimens have been prospectively validated in clinical trials or are purely model-based predictions.
Line 214–216:
The authors acknowledge the lack of verification of predicted tissue concentrations, which is a critical point. It may be valuable to suggest potential approaches or studies to bridge this gap, such as microdialysis or imaging techniques.
Line 334–366 (Future Perspectives):
Excellent discussion on the integration of imaging and virtual twins for validating PBPK models. The authors might consider briefly addressing regulatory perspectives on these innovations, especially from FDA or EMA, which could strengthen the translational aspect.
Reviewer 2 Report
Comments and Suggestions for Authors
The authors have nicely summarized the existing PBPK studies carrying out different objectives of antibiotic therapy in patients.
Overall, there is a smooth flow of information in the introduction and the discussion
I have a few comments to make
- Please justify why databases other than PubMed were not included.
- Kindly delete them – L84-86 (redundant information); L124-129 (information mentioned in the PRISMA flow chart); L159-161 (information mentioned in the table)
- Correct L161 – Efficacy is to be tested in an RCT. Models are just predicting the dose, to apply clinically for MIPD, it has to be validated.
- L342-347 – Add references
- Discuss the limitations of PBPK modeling
- Discuss if any PBPK models in externally validated even outside the antibiotics space.
- L348 – Explain how PBPK works well with sparse data.
Reviewer 3 Report
Comments and Suggestions for Authors
In line 39 and 40, please provide 1-2 examples for BW-based dose adjustment is not good enough for pediatric patients. Is it mainly due to PK difference or change of PD?
In line 82-84, please provide detailed difference of mentioned parameters in values used in pediatric patients and how different they are from adults.
For table 1, it would be informative to include current pediatric dosing regiment for listed drugs, the adult dosing regiment and limitation observed in clinics in the table to strengthen the utility of PBPK model for pediatric dose selection.
For section 3.3, please summarize the trend observed for pediatric dosing regiment and comment on the rationale on why younger patients seems to need a higher dose. From line 47-49 in background, the reduced expression or activity of drug transporters and enzymes would result in lower Cl and thus, lower dose for pediatric patients.
In table 1-Trimethoprim dosing regiment, please rephrase it for clarity.
In line 246-248, please rephrase this statement for clarity. Can the dosage be tailored with renal function? If there is limitation, you cannot make the conclusion that the developed model based on different GFR is adequate.
Please update the title of Table 3.
Reviewer 4 Report
Comments and Suggestions for Authors
The comments are mentioned in the following attachment

Round 2
Reviewer 3 Report
Comments and Suggestions for Authors
Thanks for the revision. All my comments have been addressed.